# Mycorrhizal fungi reduce the photosystem damage caused by drought stress on *Paris polyphylla* var. *yunnanensis*

**Can Huang**[1,2,3], **Xiahong He**[4], **Rui Shi**[4], **Shuhui Zi**[1,2], **Congfang Xi**[1], **Xiaoxian Li**[5], **Tao Liu**[1,2]*

**1** Yunnan Agricultural University, Kunming, China, **2** State Key Laboratory of Genetic Resources and Evolution, Kunming Institute of Zoology, Chinese Academy of Sciences, Kunming, China, **3** Guangxi Subtropical Crops Research Institute, Nanning, China, **4** Southwest Forestry University, Kunming, China, **5** Kunming Institute of Botany, Chinese Academy of Sciences, Kunming, China

\* 52133490@qq.com

**Data Availability Statement:** All relevant data are within the manuscript and its Supporting Information files.

## Abstract

Drought stress (DS) is one of the important abiotic stresses facing cash crops today. Drought can reduce plant growth and development, inhibit photosynthesis, and thus reduce plant yield. In this experiment, we investigated the protective mechanism of AMF on plant photosynthetic system by inoculating *Paris polyphylla* var. *yunnanensis*(*P.py*) with a clumping mycorrhizal fungus (AMF) under drought conditions. The drought environment was maintained by weighing AMF plants and non-AMF plants. The relative water content (RWC) of plant leaves was measured to determine its drought effect. DS decreased the RWC of plants, but AMF was able to increase the RWC of plants. chlorophyll a fluorescence curve measurements revealed that DS increased the OKJIP curve of plants, but AMF was able to reduce this trend, indicating that AMF increased the light absorption capacity of plants. DS also caused a decrease in plant Y(I) and Y(II). ETRI and ETRII, and increased Y(NO) and Y(NA) in plants, indicating that DS caused photosystem damage in plants. For the same host, different AMFs did not help to the same extent, but all AMFs were able to help plants reduce this damage and contribute to the increase of plant photosynthesis under normal water conditions.

## Introduction

Drought stress is an important abiotic factor affecting plant growth and development. With the changes in human activities drought stress has become one of the most common abiotic stresses that limit plant growth [1, 2]. Drought can have many negative effects on plants, such as damage to the structure of plant cells, which causes loss of cell function and integrity, and even the activity of cytosol and organelles can be reduced or denatured [3], causing disruption of plant metabolic systems and altering the synthesis and accumulation of plant secondary metabolites [4]. Drought also causes reduced nutrient uptake and lower growth rates in plants and affects their

**Funding:** This work was supported by the Key R & D program of Yunnan Province, China (grant no. 202103AC100003; 202101AS070228); Major special projects of the Ministry of science and technology (2021YFD1000202); the National Nature Science Foundation of China (Grant No. 31860075); the Project of the Service Center for Overseas Professionals of Yunnan Association for Science and Technology(Grant No. SW202300023);Key projects of Yunnan provincial natural fund(202201AS070064).

**Competing interests:** The authors have declared that no competing interests exist.

**Abbreviations:** Y(I), Quantum yield of PSI; Y(II), Quantum yield of PSII; Y(NA), Quantum yield of non-photochemical energy dissipation due to acceptor-side limitation; Y(ND), Quantum yield of non-photochemical energy dissipation due to donor-side limitation; Y(NO), Yield of non-regulated energy dissipation.; Y(NPQ), Yield of regulated energy dissipation; ETRI, PSI Electron transport; ETRII, PSII Electron transport.

photosynthetic rates, negatively impacting the photosystem [5]. Also, severe drought stress can lead to accelerated leaf senescence by limiting $CO_2$ diffusion to chloroplasts [6].

Photosynthesis is an essential physiological step for plant growth, and plants perform photosynthesis to maintain their energy supply. However, the interaction between photosynthesis and drought stress is complex, and a large number of studies on plants and drought stress have focused on photosynthesis. Plant photosynthetic responses are very sensitive to drought [7], and drought can contribute to increased photorespiration, increased mitochondrial respiration, Rubisco inactivation, reduced photosystem II (PSII) activity, and impaired cystoid membranes, which in turn leads to reduced ATP synthesis and increased permeability of cystoid membranes to $H^+$ ions [8]. Secondly, persistent drought also decreases chlorophyll content in the leaves, causing a decrease in the absorbance of the leaves and further reducing photosynthesis in plants [9].

Arbuscular mycorrhiza fungi (AMF) are common partners of terrestrial plants and angiosperms are the largest mycorrhizal species, with about 85% of angiosperms able to have a symbiotic relationship with AMF [10]. AMF plant mycelium is able to attach to plant roots and the mycelium continues to penetrate the root epidermis to reach the plant root cells, where it forms a clumping structure capable of nutrient exchange with the plant [11]. AMF can help plants to obtain more water and mineral nutrients, such as N and P, to promote plant growth and development [12], and can also help plants to resist the negative effects of different abiotic stresses through the exchange of nutrients [13]. It has been shown that AMF can alter water channel protein activity [14] and affect the electron allocation in the respiration rate of root cells [15], and through this interaction, AMF can effectively help plants cope with drought stress.

The aim of this study was to investigate the changes of photosynthetic system between AMF plants and non-AMF plants under drought stress, including the quantum yield of PS II and PS I and the photoprotective effect of AMF on *Paris polyphylla* var. *yunnanensis*(*P.py*). *P. py* which belongs to *Liliaceae Juss*. has great medicinal value [16]. Wild *P.py* generally grows in broad-leaved forests between 1400 and 3100 m and prefers moist and shaded environments. Photosynthesis is one of the key processes affected by drought, capable of less CO2 diffusion to chloroplasts and metabolism limiting plant growth and development [17]. It was found that drought significantly reduced the activity of the electron transport chain between maize photosystem II (PSII), photosystem I (PSI), and PSII and PSI by inhibiting electron transport from the donor side of PSII to the terminal electron acceptor of PSI [18]. While shade plants should be more sensitive to changes in photosynthesis in the face of drought stress, most current studies have focused on the interaction between drought and photosynthesis, as well as the interaction between drought and AMF, and there are few studies on the interaction between the three [14, 17, 18]. Therefore, it is necessary to study the relationship between *P.py*, AMF, and photosynthesis.

## Materials and methods

### Planting and AMF colonization of *P.py*

*P.py* was used as the plant material for this experiment. Two-year old seedlings of *P.py* with consistent growth were selected and planted in pots at the same time in May 2020. The soil was red soil: nutrient soil = 1:3. 5 *P.py* seedlings were planted with 1 kg of soil mixture.

### Drought treatment

The germination of DS *P.py* began successively in April and May 2022. To ensure that the germination of *P.py* was followed by drought treatment, *P.py* was first fully irrigated to ensure that the soil water content was maximized. The drought gradient was then determined by

measuring the soil water content. The soil water content was 15∼18% under drought stress and 33∼35% under normal wet conditions. A total of 90 days were treated. Soil moisture was maintained using a moisture meter and weighing method. The AM fungus *Glomus eburneum* (*G.e*) and *Paraglomus occultum* (*P.o*) were obtained from the Institute Mycorrhiza of Plant Nutrition and Resources, Beijing Academy of Agriculture and Forestry. The experimental treatments were as follows: plant grown in normal moisture soil with non-AMF (WCK), plant grown in normal moisture with *G.e* (WGe), plant grown in normal moisture with *P.o* (WPo), plant grown under drought stress with non-AMF(DCK), plant grown under drought stress with *G.e* (DGe), plant grown under drought stress with *P.o* (DPo),

## Amplification and Colonization of AMF

Firstly, the purchased AM spores were inoculated in the soil around the maize seeds and cultured normally for 3–6 months. Then collect the roots and surrounding soil of the maize plants that have been expanded, cut and mixed, and evenly placed in the soil around the rhizome of *P.py*.

## Determination of colonization rate

The colonization rate of AMF-inoculated roots was determined by the method of Koske RE et al. [19]. The roots of the AMF-inoculated *P.py* were first collected, washed with water and then uniformly cut into 1 cm size root segments. The root segments were soaked in 10% NaOH and placed in a 90°C water bath for 1 h. The roots were then washed with water and acidified with 2% HCL for 5 min. Finally, the roots were stained with 0.05% Taipan Blue and glycerol lactate solution for 6 h. The obtained roots were ready for microscopic observation.

## Relative Water Content (RWC)

Leaf RWC and oil RWC was determined with reference to the method of Turner NC [20].

## Determination of chlorophyll content

According to the method of Sartory et al. [21] for the determination of photosynthetic pigment content of *P.py*, fresh plant leaves of uniform length were selected, the midvein was removed, cut and weighed to 0.1 g. The samples were put into a mortar with 2–3 ml of 95% ethanol and a small amount of calcium carbonate to make a homogenous slurry, 95% ethanol was added dropwise until the tissue turned white, left to stand, filter paper was placed on a funnel, moistened with ethanol The filter paper was placed on a funnel, moistened with ethanol, and filtered into a 25 ml volumetric flask. The chlorophyll on the filter paper was rinsed with 95% ethanol into a volumetric flask, and the volume was fixed to 25 ml, shaken well, and set aside. The above sample was taken into a cuvette and measured at 665 nm, 649 nm and 470 nm with 95% ethanol reagent as blank.

Calculation formula.

$$Ca = 13.95 * A665 - 6.88 * A649$$

$$Cb = 24.96 * A649 - 7.32 * A665$$

$$Cx = (1000\ A470 - 2.05\ Ca - 114.8\ Cb)/245$$

Content (mg/g) = [C(mg/L) × total extract (25 ml)]/[weight of leaf sample (0.1 g) × 1000]

## Fluorescence dynamics

The parameters related to the daily variation of chlorophyll fluorescence in the leaves of *P.py* were determined using a PAM-100 chlorophyll fluorometer [22]. After the fluorescence signal (F′) was at a relatively stable level, the initial fluorescence (Fo′), the maximum fluorescence (Fm′), and the maximum quantum efficiency under PSII light Fv′/Fm′ = (Fm′ - Fo′)/Fm′ of chlorophyll were measured, and the initial fluorescence (Fo), the maximum fluorescence (Fm), and the maximum fluorescence (Fm) under dark adaptation were measured after dark adaptation of *P.py* leaves with dark adaptation clamps for 30 min. Fm), and the maximum quantum efficiency Fv/Fm = (Fm—Fo)/Fm under dark adaptation were measured.

The fluorescence kinetics of the leaves were measured using Dual-PAM-100 at the center of the plant leaves. Ten measurements were performed for each replicate. The plants were allowed to acclimatize in the dark for 30 min before the measurements. PSII chlorophyll fluorescence was measured using the "Fast Acquisition" mode. The leaves were first induced with 349 μmolm$^{-2}$s$^{-1}$ of saturating light, and the chlorophyll fluorescence signal was recorded starting at 10 ms and ending at 1 s. The OJIP curves were analyzed according to the method of Strasser [23]. Measurement of chlorophyll fluorescence induction curves. In the "SP-Analysis" mode, the light intensity [μmol(photons)m$^{-2}$s$^{-1}$] gradient was set to 349, and the minimum fluorescence of PSII under light acclimation was recorded. The minimum fluorescence (Fo′) and maximum fluorescence (Fm′) of PSII and the maximum fluorescence signal (Pm′) of PSI reaction center P700 were recorded under light adaptation. The fluorescence parameters were calculated by referring to the methods of Huang [24].

$$Wk = (Fk\ Fo)/(Fj\ Fo)$$

$$Vj = (Fj\ Fo)/(Fm\ Fo)$$

where Fo, Fk, Fj and Fm represent the fluorescence values at 20 μs, 300 μs, 2 ms and 300 ms, respectively, Wk is the relative variable fluorescence of K-phase, and Vj is the ratio of variable fluorescence Fj to Fo-Fp amplitude.

## Energy conversion measurements in PSI and PSII

Quantum yields of energy conversion in PSI and PSII were measured on intact leaves of *P.py* by the saturation pulse technique using the pulse amplitude modulation system Dual-PAM-100. Plants were dark-adapted at 25˚C ± 2˚C for 30 min prior to measurement. The induction curves were recorded with SP for 5 min (one pulse every 15 s). The induction curves recorded by the Dual-PAM-100 software allow the calculation of the effective quantum yields of Y(I) and Y(II) corresponding to the energy of photochemical conversion in PSI and PSII. Then, Y(ND), Y(NA), Y(NPQ) and Y(NO) are calculated from the quantum yields dissipated in this process. In the "SP-Analysis" mode, ETR(I) and ETR(II) are recorded by the software.

## Data analysis

Data were analyzed using spss25 software. The results were analyzed by one-way ANOVA. Significance was determined at $P < 0.05$, and results were expressed as mean and standard deviation. All analyses were performed in three replicates. Plots were made with origin2021 software.

## Result

### Effect of drought stress on AMF colonization rate

Under different moisture conditions, the roots of *P.py* inoculated with AMF treatment were infested with mycelium and all could see mycelium or spore structures(S1 Fig); the treatments without inoculation had almost no mycorrhizal structures seen in either normal moisture or moderate drought. Under normal moisture treatment, the colonization rate of AMF was significantly higher than that of drought treatment, in which the colonization rate of Ge under normal moisture treatment increased by 36% compared with that of drought treatment, while Po increased by 19%. It can be seen that drought can reduce the symbiosis of AMF with plants Fig 1) (P<0.05).

### Relative soil water content and chlorophyll content

The drought treatment significantly reduced the leaf and soil RWC compared to the control, but inoculation with AMF alleviated this trend (Table 1). It can be found that inoculation of AMF appeared to increase the soil RWC, but not significantly. Under drought stress, the chlorophyll a content of *P.py* decreased significantly, especially in plants not inoculated with AMF, while after inoculation with AMF, *P.py* slowed down the decreasing trend of chlorophyll a caused by drought, and under normal moisture, AMF also increased the chlorophyll a content of *P.py*, where the highest chlorophyll a content was inoculated with Ge, reaching 1.96 mg/g, while the The chlorophyll a content of plants not inoculated with AMF under normal water was 1.69 mg/g, while under drought stress, inoculation with Po helped the accumulation of chlorophyll a content of *P.py* the most, reaching 1.6 mg/g, which was significantly higher than that of DCK and DGe (Fig 2). The contents of chlorophyll b and carotenoids were consistent with those of chlorophyll a. Moreover, under drought stress, AMF was significantly higher than chlorophyll a for the increase of chlorophyll b and carotenoid contents, and the

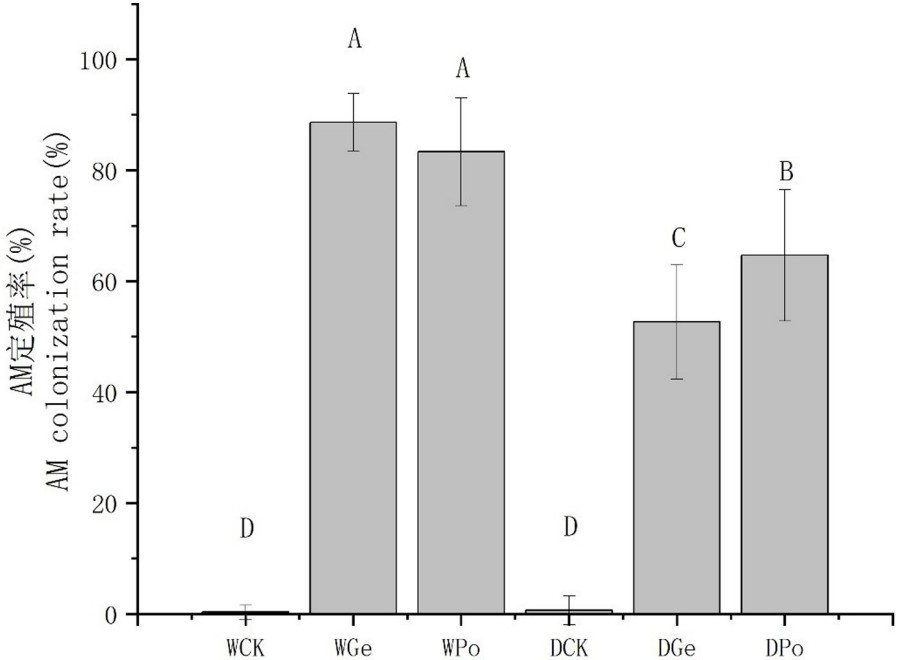

**Fig 1. Response of plants with different treatments to drought stress and AMF colonization.**

**Table 1. Effect of drought and AMF on the relative water content of plants and soil in different treatments.**

| treatment | RWC(leaf) % | RWC(soil) % |
|-----------|-------------|-------------|
| WCK | 86 | 65 |
| WGe | 88 | 66 |
| WPo | 81 | 68 |
| DCK | 43 | 34 |
| DGe | 57 | 37 |
| DPo | 48 | 36 |

chlorophyll b and carotenoid contents of *P.py* inoculated with AMF were equal to or even higher than those of plants treated with normal water without AMF (Fig 2).

## Effect of drought stress on gas exchange parameters of *P.py*

Under adequate water conditions, inoculation with AMF significantly increased the photosynthetic rate, stomatal conductance, and transpiration rate of *P.py* (Table 2). Under drought treatment, the photosynthetic rate, stomatal conductance, and transpiration rate of *P.py* were significantly reduced, but this trend was mitigated by inoculation with AMF, and the net photosynthetic rate and transpiration rate of *P.py* were significantly increased after inoculation with Ge, with the photosynthetic rate increasing by 0.9 and the transpiration rate by 0.36, while inoculation with Po increased the intercellular $CO_2$ concentration of the plant. It can be shown that AMF inoculation can have a positive effect on the photosynthetic rate of plants.

## Fast fluorescence kinetic parameters

In this study, we found that drought stress was able to significantly attenuate the Fv/Fm values of *P.py* by measuring the changes in Fv/Fm (Table 3), but this phenomenon was greatly alleviated after inoculation with AMF, indicating that AMF did slow down the damage of the plants by drought stress. In addition, we found that drought caused a significant reduction in plant PIABS (Table 3), which is related to chlorophyll reaction centers and PSII primary photochemistry, indicating that drought also reduced the photoconversion efficiency of *P.py*, and this trend was found to be mitigated only in inoculated Ge plants.

The increase in Fo indicates a decrease in the energy transferred to the PSII reaction center, which may be related to the photochemical damage to PSII, and the decrease in the rate of increase in Fo after inoculation with AMF can indicate a decrease in its damage. AMF makes the increase in the efficiency of the PSII donor-side complex, which can effectively protect the oxygen release complex and enhance the photochemical effect of PSII. In addition, the chlorophyll fluorescence kinetic curve (O-K-J-I-P curve) (Fig 3) can also reflect the damage to the donor and acceptor sides of PSII in plants. The change in the J phase of the curve can also indicate the magnitude of the central electron transfer capacity from QA to QB. In this study, drought stress caused the relative fluorescence ratio of K point (Fig 3) and J point relative fluorescence to be higher than that of normal water plants, which could indicate that the plant PS II donor side oxygen release complex (OEC) was damaged at this time. It was shown that the O-J-I-P curve Wk values of plants increased and OEC activity was inhibited after drought treatment, which was similar to the present study. In general, drought stress can lead to photoinhibition in *P.py*, and its light capture, conversion and absorption were slowed down to different degrees, and inoculation with AMF could alleviate this trend to some extent.

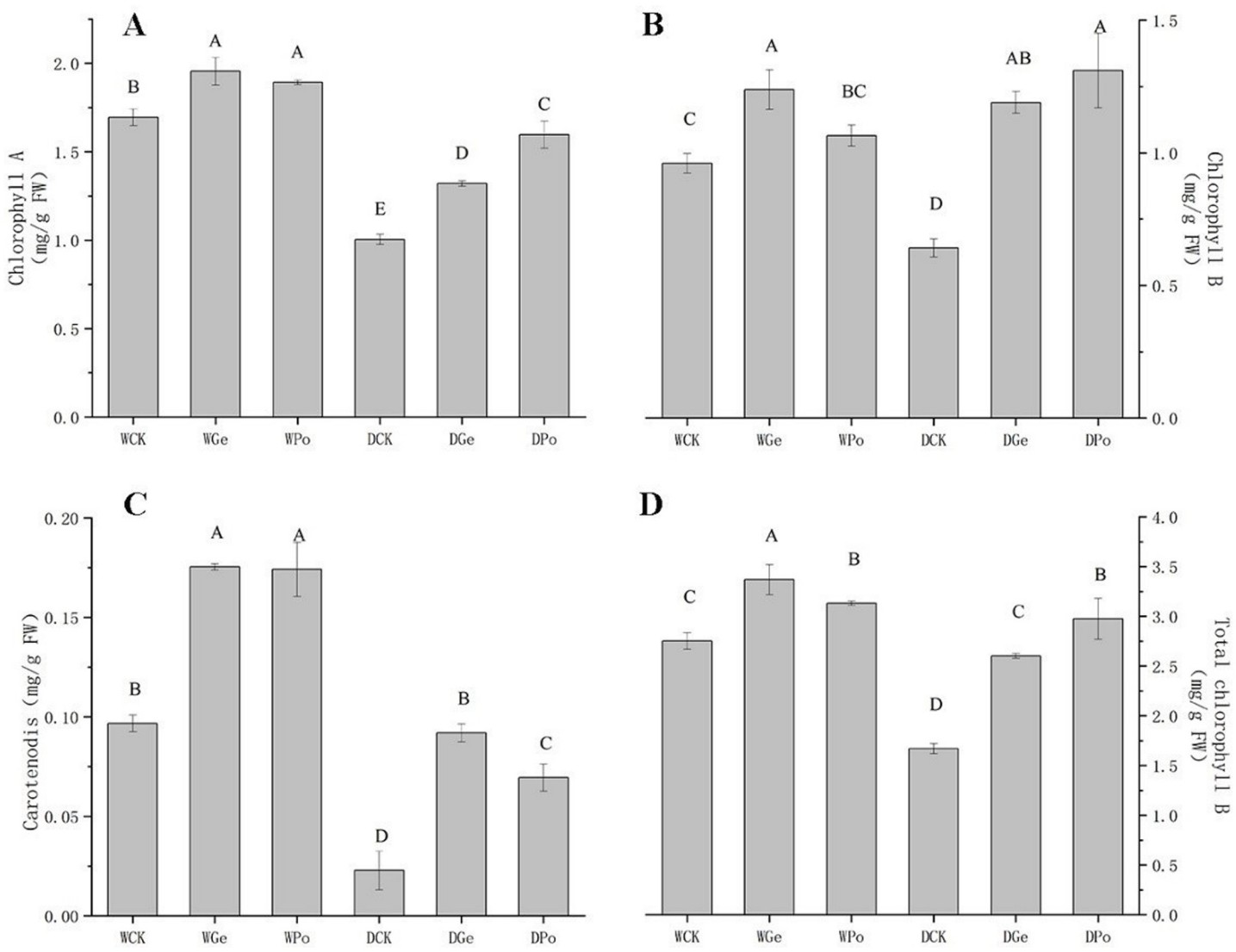

**Fig 2. Effect of drought and AMF on the chlorophyll content of plants with different treatments.**

**Table 2. Effect of drought and AMF on gas exchange parameters of plants in different treatments.**

| Treatment | Photo | Cond | Ci | Trmmol |
|---|---|---|---|---|
| WCK | 2.87±0.173B | 0.058±0.002B | 314.95±7.300A | 0.73±0.028B |
| WGe | 2.45±0.163B | 0.085±0.019A | 324.28±18.830A | 1.07±0.212A |
| WPo | 3.13±0.430A | 0.101±0.036A | 340.22±14.901A | 1.21±0.363A |
| DCK | 0.38±0.099D | 0.018±0.002B | 265.24±70.021B | 0.25±0.028D |
| DGe | 1.22±0.118C | 0.050±0.004B | 248.59±15.799B | 0.61±0.058B |
| DPo | 1.20±0.111D | 0.028±0.005B | 318.83±2.863A | 0.38±0.062C |

Each experiment was repeated thrice. Significance was determined at P < 0.05and the results are expressed as mean values and standard deviation (SD)

**Table 3. Various Chl a transient parameters in different treatments.**

| Treatment | F$_O$ | Fm | Fv/Fm | Fv/Fo | Vj | PI $_{(abs)}$ |
|-----------|-------|-----|-------|-------|-----|---------------|
| WCK | 0.37±0.036BC | 1.22±0.150AB | 0.39±0.050BC | 2.33±0.293AB | 0.47±0.032C | 8.45±0.840A |
| WGe | 0.35±0.006C | 0.86±0.005C | 0.66±0.044A | 2.73±0.105A | 0.49±0.026C | 6.28±1.050B |
| WPo | 0.38±0.031BC | 1.07±0.018B | 0.48±0.026B | 2.08±0.326BC | 0.52±0.019BC | 6.34±0.390B |
| DCK | 0.43±0.094ABC | 1.30±0.069A | 0.38±0.002C | 0.75±0.482E | 0.56±0.036A | 5.66±1.160B |
| DGe | 0.45±0.043AB | 1.24±0.123AB | 0.45±0.048BC | 1.50±0.117D | 0.55±0.010AB | 5.64±0.372B |
| DPo | 0.50±0.034A | 1.26±0.177AB | 0.45±0.476BC | 1.75±0.100CD | 0.56±0.021AB | 5.68±0.031B |

Each experiment was repeated thrice. Significance was determined at $P < 0.05$ and the results are expressed as mean values and standard deviation (SD)

## Effect of drought stress on quantum yield of PSI and PSII

By studying the quantum yield of PSI and PSII, for PSII, Y(II) indicates the effective photo-chemical quantum yield in PSII. And the quantum yield lost in this process was divided into Y

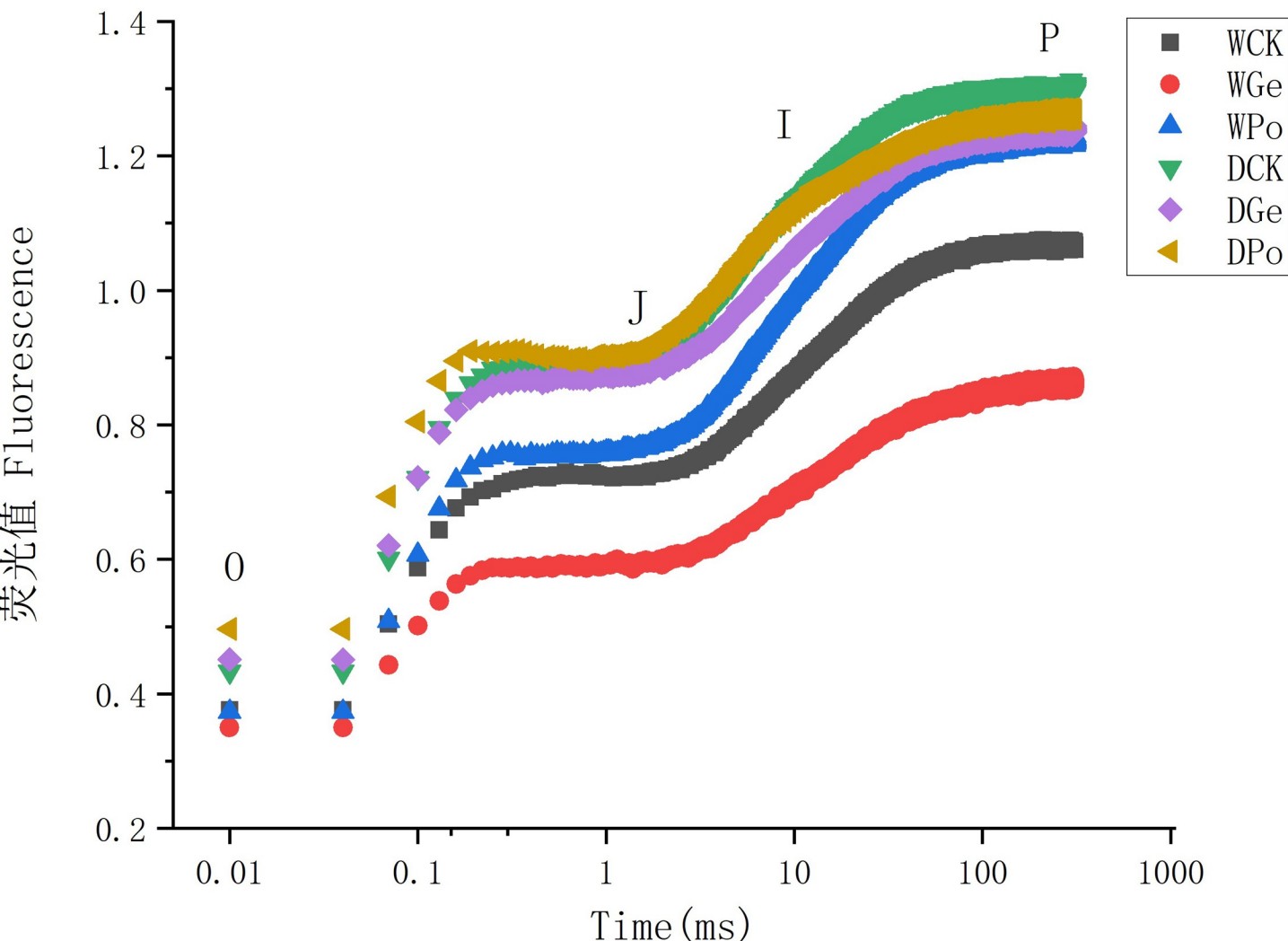

**Fig 3. Chl a transient curve 'in different treatments.** Each experiment was repeated thrice.

(NPQ) and Y(NO), where Y(NPQ) reflects the high or low share of leaf dissipation into thermal energy through regulated non-photochemical burst mechanism, while Y(NO) is related to the degree of plant photodamage. In contrast, ETR indicates the response of leaves to photosynthetic electron transfer. In addition, Y(ND) is the non-photochemical quantum yield due to PSI donor-side limitation, and Y(NA) indicates the non-photochemical quantum yield due to PSI acceptor-side limitation.

First, inoculation with AMF was able to increase Y(II) of *P.py* under water-sufficient conditions, with the inoculation of Eleutherococcus balsamifera enhancing Y(II) the most (Fig 4A). Under drought stress, the Y(II) of the plants was significantly reduced, indicating that the

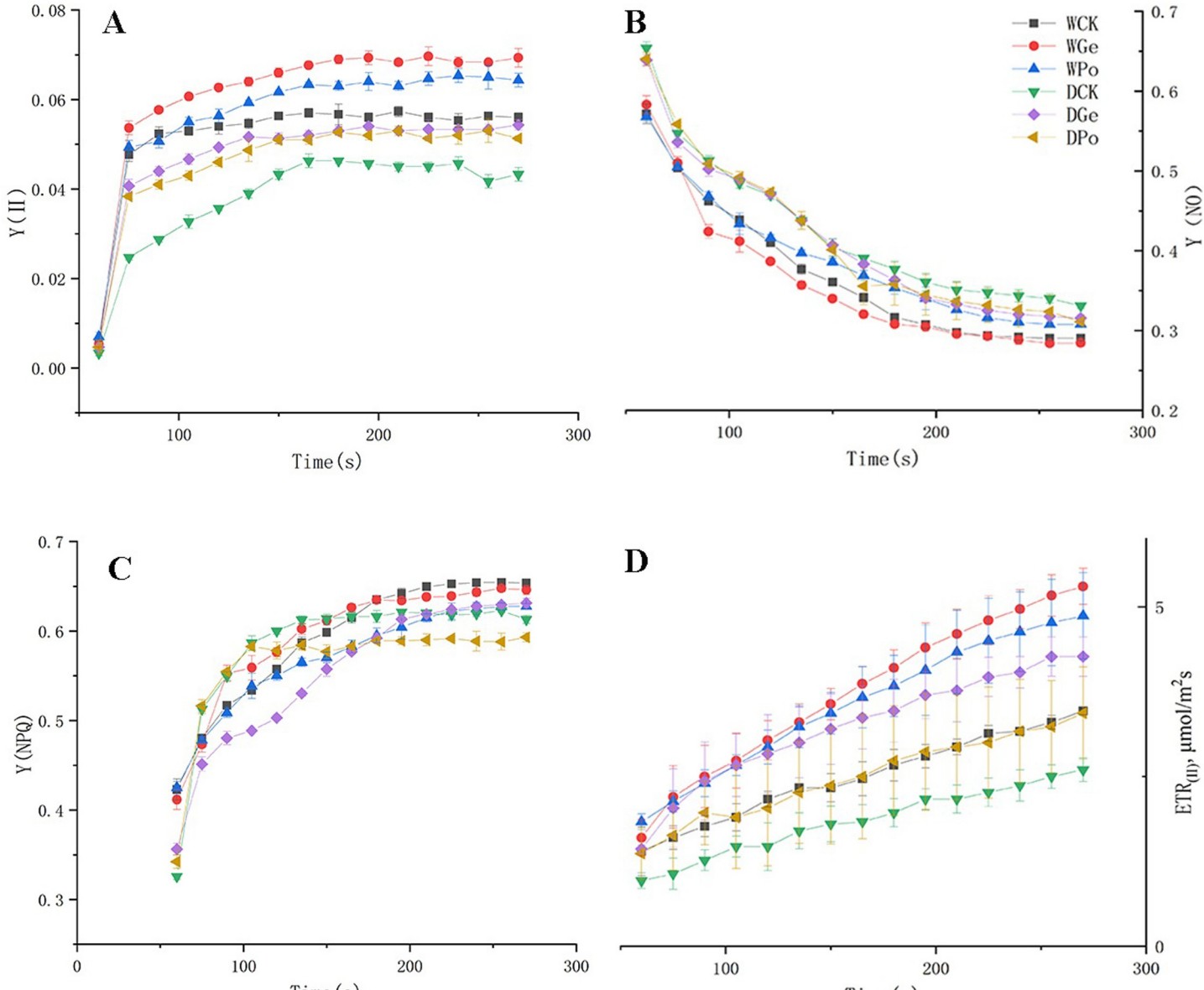

**Fig 4.** Effect of drought stress and AMF colonization in different treatments on the quantum yields of energy conversion in PSII where a Y(II) is the quantum yield of PSII, b Y(NO) is the yield of non-regulated energy dissipation, c Y(NPQ) is the yield of regulated energy dissipation, d ETRII relative electron transport rates in PSII with the application of a saturation pulse.

energy allocated to Y(II) after light energy absorption by the plants decreased, but after the symbiosis of *P.py* and AMF, both showed a trend of increasing Y(II) energy. And the energy of Y(NPQ) and Y(NO) also stabilized after a significant increase (Fig 4B and 4C). In leaf electron transport, ETRII appeared at a higher rate in AMF plants. On the other hand, under drought stress (Fig 4D), Y(I) was significantly decreased in *P.py* (Fig 5A), and Y(NA) and Y(ND) were significantly enhanced (Fig 5B and 5C), but inoculation with AMF was able to significantly slow down this trend, while ETRII likewise appeared at a higher rate in AMF plants. It can be shown that AMF played a positive role in protecting the plants from drought damage, and among the two AMFs, Ge was more helpful in resisting environmental stresses in *P.py*.

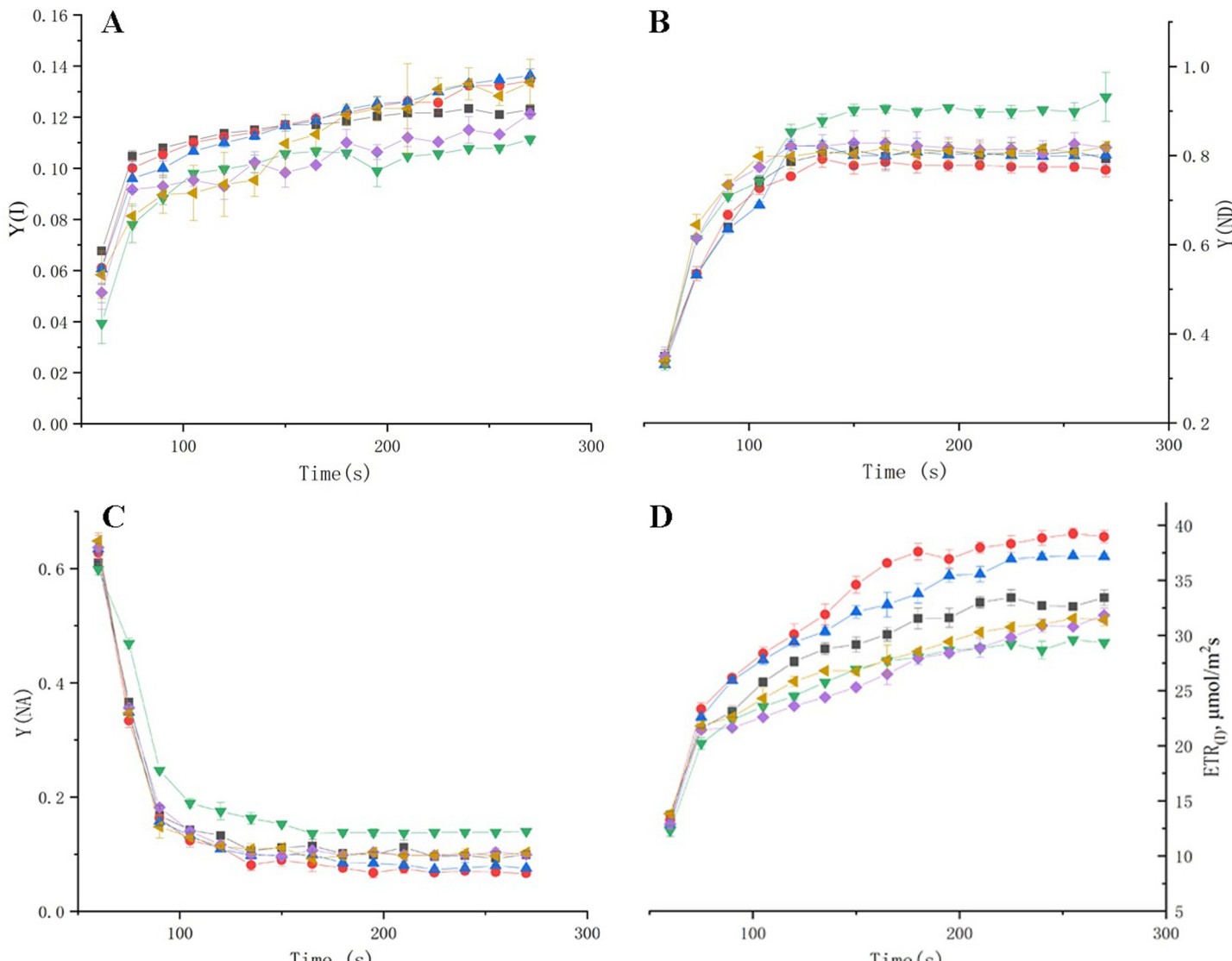

**Fig 5.** Effect of drought stress and AMF colonization in different treatments on the quantum yields of energy conversion in PSI where a Y(I) is the quantum yield of PSI, b Y(NA) is the quantum yield of non-photochemical energy dissipation caused by acceptor-side limitation, c Y(ND) is the quantum yield of non-photochemical energy dissipation caused by donorside limitation, d ETRI relative electron transport rates in PSI with the application of a saturation pulse.

### Effect of DS on P700 Redox kinetics

The magnitude of Pm can represent the amount of effective PSI complexes and can also indicate the degree of loss produced by the plant in response to environmental stresses. Drought stress was able to reduce the Pm,Pm' of the plant (Fig 6). In contrast, the non-water-stressed plants in symbiosis with AMF reached the highest Pm,Pm' values, showing that AMF can play a protective role for the plants. And AMF was also able to increase the Pm,Pm' of the plants under normal water treatment.

## Discussion

The symbiotic relationship between terrestrial plants and AMF is very common, and the two achieve mutual benefits by exchanging nutrients. In this experiment, after the occurrence of drought stress, both AMF colonization rates showed a significant reduction, but the magnitude of the reduction still differed between the two, and the colonization rate of Po was significantly higher than that of Ge under drought conditions, which was significantly different from that of the two AMF colonization rates under normal moisture treatment. This indicates that drought stress can reduce the colonization rates of Ge and Po with *P.py*, and studies have shown that drought can strongly inhibit the AMF colonization rates of both primary and lateral roots of plants, which is similar to the present study [25]. There are still few studies related to whether AMF can maintain the colonization rate of plants under non-stressful environments when they are exposed to stress, and some studies have shown that drought stress can promote a symbiotic relationship between plants and AMF [2], which is contrary to the conclusion of the present study that although most AMF can help plants grow in adversity, when

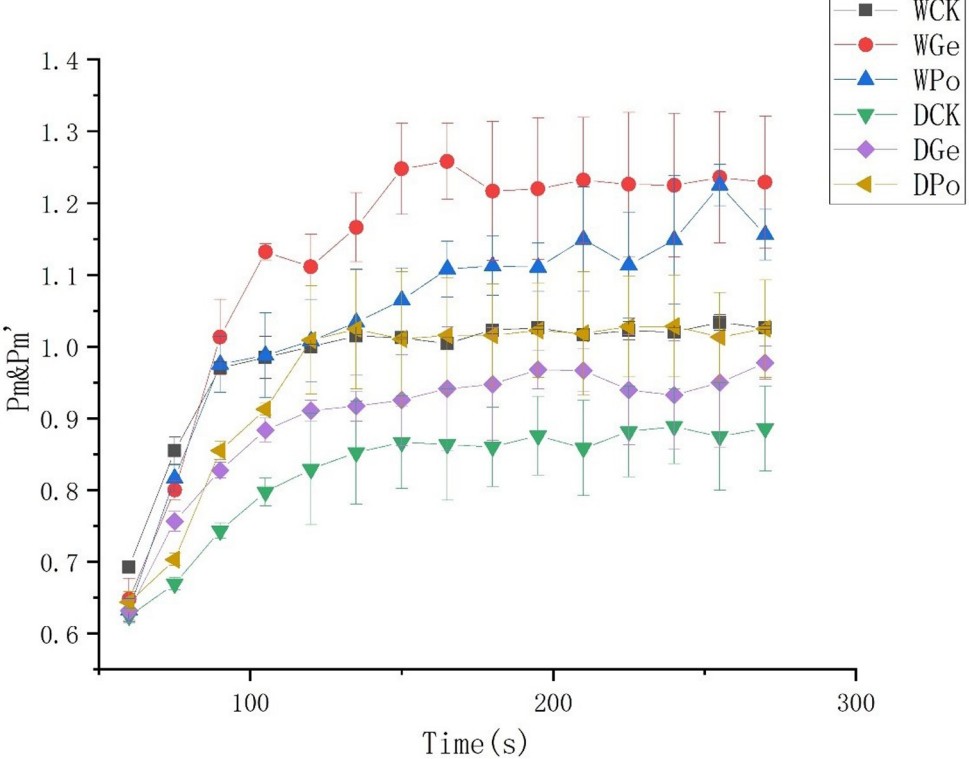

**Fig 6. The response of Pm (the maximal change in the P700 signal) and Pm' (the maximal change in the P700 signal in a given light) in different treatments plants with the application of a saturation pulse.**

co-resisting environmental stresses with plants, different AMF may exhibit different levels of stress resistance and stress response, and *Glomeraceae* class AMFs are able to better symbiosis with plants by reducing carbon mobility, for example [26], which may be one of the reasons for their reduced colonization rate.

## AMF increase plant photosynthesis under drought conditions

Plant photosynthesis is the main way of energy production in plants, and previous studies found that AMF inoculation of plants significantly enhanced plant photosynthetic rate and alleviated the constraints of stomatal factors brought about by drought, and AMF also increased the accumulation of photosynthetic pigments in plants, which is consistent with the findings of this paper [5]. It was shown that AMF was able to alter the expression of plant transmembrane proteins and photosynthesis-related electron transport enzymes up-regulated, while AMF also improved plant P-nutrient uptake and regulated antioxidant systems and osmotic pressure, thereby improving plant photosynthesis [27, 28]. Under adequate water conditions, inoculation of AMF significantly increased photosynthetic rate, stomatal conductance, and transpiration rate of *P.py*, which is consistent with previous studies [29]. Under moderate drought conditions, plants showed a significant decrease in photosynthetic rate, stomatal conductance, and transpiration rate, but AMF was able to slow down this trend, and in the assay of photosynthetic pigments, it was found that inoculation with AMF maintained chlorophyll a, b, and carotenoids that were reduced in plants due to drought stress.

Leaf water content is an important indicator of plant exposure to drought stress, and drought stress in wheat leaves leads to a decrease in leaf water content, resulting in stomatal closure and growth arrest [30]. However, AMF inoculation contributed to the increase in water content of drought plants [31], a result similar to our experiments. AMF can enhance the water use efficiency of plants, as well as alter the content of osmotic substances in plant cells, thus helping plants to resist drought stress [32]. Drought stress can lead to a decrease in photosynthetic rate and a decrease in stomatal conductance, which can trigger plant death [33, 34], and plants reduce their nutrient uptake through stomatal regulation under drought conditions, resulting in a decrease in water and nutrients in the plant. Our experiments proved the conclusion of previous experiments [1] that inoculation of AMF can effectively suppress the effects of drought stress, and plants can improve their photosynthetic rate, stomatal conductance, and transpiration rate through AMF symbiosis enough to protect their photosynthetic system from drought loss, thus better helping plants to carry out normal physiological cycles.

The effect of drought on the plant can be quickly and obviously reflected in the photosynthetic system, and a large number of studies have been conducted to more accurately study the negative effects of drought by analyzing the photosynthetic system of plants after drought [35], and OJIP can accurately respond to changes in electron transfer in the photosynthetic system of plants through changes in the fluorescence curve. o-phase generally represents the initial state of electron acceptance on the receptor side of PS II and can respond to the plant ability to absorb light [36]. When plants are subjected to drought stress, the O point usually shows a tendency to increase, and there is also a clear K peak [37], which can indicate that damage to the oxygen release complex has occurred [38]. This is similar to the conclusion of the present study that drought stress can lead to a rise in OKJ, and that drought inhibition of extra-QA electron transport may be the main reason for the rise in J-phase, suggesting that drought stress negatively affects extra-QA electron transport, while AMF inoculation alleviates this trend (Fig 3) Significant changes were also observed among different treatments in the J-P phase, and the rise in I-P was correlated with the reduction of P700+ in plastocyanin Pc and PSI [39], and possibly also with the number of PSI reaction centers [40], and drought stress

significantly increased the I-P phase of plants, indicating that both the donor and acceptor sides of PSII in *P.py* leaves were damaged under drought treatment. Drought stress was able to cause photoinhibition in *P.py*, and its light capture, conversion and uptake were slowed down to varying degrees, while inoculation with AMF was able to alleviate this trend again to some extent, and AMF helped the plant to absorb more water and nutrients may be the main reason for this phenomenon. Drought can damage the plant photosystem, which can be quickly and effectively studied by measuring the fast fluorescence-induced kinetic curves. Fv/Fm is an indicator of the degree of stress on the plant, and studies have shown [41] that the changes in Fv/Fm are small in normal environments and the effects of different species and growth conditions are small. However, when plants are subjected to drought stress, there is a significant decrease in Fv/Fm. In addition, water deprivation may also trigger salinity stress, which is one of the factors contributing to the reduction of Fv/Fm [42]. In this study, by measuring the changes in Fv/Fm, we found that drought stress was able to significantly reduce the Fv/Fm values of *P.py* (Table 3), but this phenomenon was greatly alleviated after inoculation with AMF, indicating that AMF did slow down the damage of the plants by drought stress. In addition, we found that drought causes a significant decrease in plant PI (ABS) (Table 3) and that PIABS is related to chlorophyll reaction centers and PSII primary photochemistry and can respond to chloroplast health [43], indicating that drought stress also decreases the photoconversion efficiency of *P.py*.

Photosystem is an important component of plant photosynthesis, and plants usually exhibit lower ETR(II) and Y(II) and higher Y(NO) and Y(NPQ), after being subjected to drought stress. Through the study, Y(II) of AMF plants were all higher than those of non-AMF-inoculated plants, and the decreasing trend was significantly smaller than that of AMF-plants after AMF plants were subjected to drought treatment. It was shown that AMF can induce the plants to enhance the assimilation effect of $CO_2$ and thus increase their Y(II) [5]. This could indicate that AMF can effectively protect Y(II) from damage, which is also indicated by Y(NO), and this increase in Y(NO) indicates that the plant absorbs excess light intensity for a certain period of time and its system may be photodamaged, resulting in the inability to safely release the excess light energy as heat, which is the main reason for the decrease in Y(II), and in the present study, AMF effectively reduced the increase of Y(NO), which is consistent with the trend of Y(II). While the increase in Y(NPQ) likewise implies damage to PSII, this series of reactions is usually affected by the occurrence of its electron transport, making a decrease in NPQ that depends on the proton gradient excitation across the vesicle membrane, which ultimately leads to the inability of the light energy to be consumed thermally or photochemically, causing damage to plant PSII, similar to the study of Mathur S [5]. Among the two AMFs, the Y(II) of *P.py* inoculated with Ge was significantly higher than Po, especially under adequate water treatment, and most of the AMFs were able to bring positive effects to plants under drought stress [44, 45], but the effects of different AMFs on plants varied greatly [46], and studies have shown that the effects of AMFs on their hosts could not be determined based on their morphology and classification [47]. In this experiment, the enhancement of photosynthesis by Ge more than Po may be specific to *P.py*, and this trend is also reflected in the accumulation of secondary metabolites in *P.py*, which was found in our previous experiments to be able to increase the content of secondary metabolites in *P.py* to a greater extent, which can protect the plant from more oxidative damage and thus enhance the photosynthesis under plant adversity.

In addition, Y(I) also showed a tendency to increase in AMF-inoculated plants dealing with drought stress, which led to a significant decrease in plant Y(I), and the inhibition of Y(I) is closely related to Y(II) [48], where damage to Y(I) is usually more persistent and not easily recovered, and the circulating electron flow near Y(I) is essential to protect Y(I) from damage.

In the present study, the decrease of Y(I) was fully consistent with the trend of ETR(I), while the electron flow from PSII to PSI also induced the accumulation of hydroxyl radicals on the receptor side, which exacerbated the photoinhibition of PSI. It was noted that the repair rate of PSII photodamage is influenced by ATP synthesis [49]. In this study, the significant decrease in ETR(II) made the rate of ATP synthesis via ETR(II) also significantly reduced, and only through the circulating electron flow of PSI, ΔpH was formed to drive ATP synthesis to repair PSII, but the dysregulation of PSII damage rate and repair rate made the repair rate of PSII slower and resulted in the photodamage of PSII. The Y(ND) of the plant indicates the quantum yield of non-photochemical energy dissipation at PS I due to donor-side limitation. If Y(ND) is high, it indicates that the plant receives excess light intensity on the one hand, and on the other hand, it indicates that the plant can still protect itself by increasing thermal dissipation, and Y(ND) is an important indicator of photoprotection [50]. While Y(NA) is the quantum yield of non-photochemical energy dissipation at PS I due to receptor side limitation. Inactivation of key enzymes of the Calvin-Benson cycle after dark adaptation also causes elevated Y(NA). elevated Y(NA) may be caused by photodamage [51]. In the present experiment, Y(NA) and Y(ND) were significantly increased under drought conditions, indicating that drought caused a blockage on the donor side of *P.py* and a decrease in light absorption efficiency, and also indicating a decrease in efficiency on the acceptor side. However, AMF plants reduced this trend, indicating that AMF protected the photosynthetic system of the plant under drought stress. In PSI, the P700+ signal may vary between a maximum (complete oxidation of P700) and a minimum level (complete reduction of P700) [52]. In general, Pm' is generally smaller than Pm, whereas after AMF inoculation, both Pm' and Pm were significantly increased, indicating greater PSI efficiency, whereas under drought, the decrease in Pm' and Pm levels indicates a decrease in PSI efficiency (Fig 4). It was shown that AMF was able to promote the production of more antioxidant enzymes in plants under drought conditions, reduce the damage of reactive oxygen species, and ensure the growth of plants by affecting their water content, which is similar to the findings of this study.

## Conclusion

By inoculating *Paris polyphylla* var. *yunnanensis* with AMF, we found that its growth and photosynthesis under drought stress were superior to those of non-AMF plants. Drought stress reduced the colonization rate of AMF, but increased the leaf water content, and AMF plants had higher photosynthetic and transpiration rates, ensuring their growth and development. By studying its PSI and PSII, it was found that drought stress reduced Y(II), Y(I) and electron transfer rate of plants, increased donor-side and acceptor-side limitation, and increased energy dissipation and reduced photoprotection mechanism of plants, and this injury was very obvious in PSII. However, the adverse effects of drought were mitigated by AMF inoculation, which was mainly caused by the ability of AMF to improve plant water use and nutrient uptake, and there were still differences between AMFs for the same host, and inoculation with Ge helped plants more than inoculation with Po under drought conditions. but the trend of AMF helping plants to improve stress resistance under drought stress was consistent, and Inoculation of AMF could better help plants to improve the stability and resistance of photosynthetic system under drought stress.

## Supporting information

**S1 Fig. Mycorrhizal structure of the roots of *Pp*.** (A structure of mycorrhizal mycorrhizal infestation by Ge under normal moisture, B structure of mycorrhizal infestation by Po under normal moisture, C structure of mycorrhizal infestation by Ge under drought stress, D

structure of mycorrhizal fungal inoculation without tufts under normal moisture).
(JPG)

## Acknowledgments

We thank Yunnan Agricultural University for supporting the acquisition of literature and the anonymous reviewers for their valuable feedback.

## Author Contributions

**Conceptualization:** Tao Liu.

**Data curation:** Can Huang, Xiahong He.

**Formal analysis:** Can Huang.

**Funding acquisition:** Tao Liu.

**Investigation:** Xiahong He, Rui Shi, Shuhui Zi, Congfang Xi, Xiaoxian Li, Tao Liu.

**Methodology:** Tao Liu.

**Writing – original draft:** Can Huang.

**Writing – review & editing:** Can Huang.

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
