## [Decision Letter · Decision Letter 0]

7 Feb 2023

PONE-D-22-33646Mycorrhizal fungi reduce the photosystem damage caused by drought stress on Paris polyphylla var. yunnanensisPLOS ONE

Dear Dr. Huang,

Thank you for submitting your manuscript to PLOS ONE. After careful consideration, we feel that it has merit but does not fully meet PLOS ONE’s publication criteria as it currently stands. Therefore, we invite you to submit a revised version of the manuscript that addresses the points raised during the review process.

We look forward to receiving your revised manuscript.

Kind regards,

Mayank Gururani

Academic Editor

PLOS ONE

Journal Requirements:

"Liu Tao conceived, designed the experiments and wrote this paper. Huangcan analyzed the data. Zi Shuhui, Xi Congfang and Li Xiaoxian Preparing plant materials All authors read and approved the final manuscript."

4. Please include a copy of Table 2 which you refer to in your text on page 6.

Reviewers' comments:

Reviewer's Responses to Questions

**Comments to the Author**

1. Is the manuscript technically sound, and do the data support the conclusions?

Reviewer #1: Partly

Reviewer #2: Yes

2. Has the statistical analysis been performed appropriately and rigorously? 

Reviewer #1: No

Reviewer #2: Yes

3. Have the authors made all data underlying the findings in their manuscript fully available?

Reviewer #1: No

Reviewer #2: Yes

4. Is the manuscript presented in an intelligible fashion and written in standard English?

Reviewer #1: No

Reviewer #2: No

5. Review Comments to the Author

Reviewer #1: This manuscript investigated the protective mechanism of AMF on plant photosynthetic systems by inoculating Paris polyphylla var. yunnanensis (P.py) with two mycorrhizal fungi (AMF) under drought stress (DS). The authors have underlined that the DS decreased the RWC and increased the OKJIP curve of the plants, and the authors claim that AMF inoculation could reverse it. This paper also reports the photosystem damage caused by the DS and shows how AMF helped to reduce this damage. The study appears to be sound, but the language is unclear in the results, and the discussion needs more relevant references to support the findings.

Comments:

• The statistical tests and significance values of different comparisons should be clarified for tables/figures to ensure that readers understand exactly what the researchers studied.

• The results need to be described well, e.g., “and the decrease in the rate of increase in Fo after inoculation with AMF can indicate a decrease in its damage.” The authors should clarify the results section to avoid confusion.

• The authors can also report the difference between the protective mechanism of Glomus eburneum (G.e) and Paraglomus occultum (P.o) in the results section and support these findings with relevant literature in the following discussion.

• In the first paragraph of the discussion, the authors try to highlight the effects of drought stress on the symbiosis of AMF and P.py; I suggest citing more studies to show how the findings of this paper relate to previous research on symbiosis.

• 2nd paragraph of discussion: Please add more references to support the claim that “AMF inoculation of plants significantly enhanced plant photosynthetic rate under drought.” These findings need more explanation (with references to specific studies) in the context of underlying molecular processes that show how AMF rescues P.py photosynthesis under drought stress.

Minor changes/Typos:

• Page 1: Please add more supportive references in addition to reference [7]

• Page 1: please add more literature in addition to reference [8] to establish the link between drought stress and photosynthesis

• Page 2, last paragraph of introduction: Change “P.py” to “Paris polyphylla var. yunnanensis(P.py)” in the first sentence

• Page 2, last paragraph of introduction: Build up the importance of explicitly using P.py for studying DS responses in AMF and non-AMF plants.

• Page 2, last paragraph of introduction: The author states that “While shade plants should be more sensitive to changes in photosynthesis in the face of drought stress, most current studies have focused on the interaction between drought and photosynthesis, as well as the interaction between drought and AMF, and there are few studies on the interaction between the three.” Please cite relevant studies that explored these interactions.

• Material and Methods, Page 2: Italicize “Paraglomus occultum (P.o)”

• Material and Methods, Page 2: Please provide references to relevant articles/websites for protocols and software discussed in materials and methods

• Material and Methods, Page 2: “resu4=lts”

• Material and Methods, Page 2: Inconsistent styles used, e.g., “determined by the method of [17]” and “method of Sartory et al. [19]”, it needs to be consistent throughout the manuscript

• Material and Methods, Page 2: inconsistent usage “O-J-I-P” or “OJIP.”

• Table & Figures: Please add detailed captions for multiple-part figures and tables

• Table & Figures: The treatment labels used in Table 1 are different “WX”, “WYL”,”DX”, and”DYL” and no description is provided

• Table & Figures: Table 2 is missing

Reviewer #2: Using Paris polyphylla and arbuscular mycorrhiza fungi (AMF), the authors examined AMF colonization rate, relative water content, chlorophyll content and photosynthesis under drought stress. The experiments were designed well and executed efficiently. Except for typos, the data is presented coherently, and the MS is well written. For researchers engaged in plant stress research, I believe the MS is valuable. It is important that the authors address the following issues, however:

1. In Abstract, Capitalize the beginning of the sentence like Chlorophyll a fluroscence curve measuresments…, DS also caused a descrease…

2. Explain YI, YII, ETRI and ETRII in abstract.

3. I found some inconsistencies, such as arbuscular mycorrhiza fungi (AMF) is written as AMF plant and AM plant throughout the MS.

4. Kindly type in bold the sub-headings and Capitalize the first letter such as Drought treatment.

5. Typo error in the drought treatment, the germination of the D P.py

6. In drought treatment, how much water did you use for normal wet condition and drought condition. Did you withdraw water completely in drought condition?

7. Include the microscopic images of colonization.

8. Typo error in Relative water content such as oil to soil.

9. In determination of chlorophyll content repeat of the sentence the filter paper was placed on a funnel, moistened with ethanol.

10. In data analysis, typo error in results were analysed

11. In table 1, explain what is WX, WYL, DX, DYL.

12. Where is table 2? Most of the table numbers are incorrectly written.

13. Expand photo, cond and ci in Table 3.

14. In results, effects of drought stress on quantum yields of PSI and PSII, inoculation of Eleutherococcus balsamifera is given. This part was not seen in materials and methods.

15. A proofreading should be done for typos.

6. PLOS authors have the option to publish the peer review history of their article (what does this mean?). If published, this will include your full peer review and any attached files.

Reviewer #1: No

Reviewer #2: No

---

## [Author Response · Author response to Decision Letter 0]

14 Aug 2023

Reply to Reviewer #1

The research background introduction of the article was added, relevant references [17,18,27,28] were added, and more information on the relationship between drought and photosynthesis was provided.

The sentence order of the last paragraph of the introduction was changed, and p.py was placed in the first sentence. The importance of this study is highlighted.

Additions were made to address the relevant measurements of PSI and PSII in the material methods, and typos and incorrect formatting were revised.

The incorrect presentation of 4=lts was modified.

The usage of OJIP curves was unified.

Changed the treatment numbers in Table 1, WX, WYL, DX, and DYL to WGe, WPo, DGe, and DPo, respectively.

Corrected the table serial numbers, the previous manuscript incorrectly labeled Table 2 as Table 3.

Reply to Reviewer #2

Added explanation of some technical terms after the abstract.

Harmonized the expressions of AMF plants and AM plants, and changed AM plants to AMF plants uniformly.

Revised the misrepresentation of D.P.py.

Drought conditions are mainly adjusted by moisture meter, according to the measurement results, the soil water content under drought and sufficient moisture conditions is controlled within the corresponding range. For example, drought soils are measured at 15-18% moisture by the moisture meter, and 33-35% for adequate moisture treatment soils.

Images of root colonization of P.py are provided in the additional images.

---

## [Decision Letter · Decision Letter 1]

30 Aug 2023

PONE-D-22-33646R1Mycorrhizal fungi reduce the photosystem damage caused by drought stress on Paris polyphylla var. yunnanensisPLOS ONE

Dear Dr. Huang,

Thank you for submitting your manuscript to PLOS ONE. After careful consideration, we feel that it has merit but does not fully meet PLOS ONE’s publication criteria as it currently stands. Therefore, we invite you to submit a revised version of the manuscript that addresses the points raised during the review process.

We look forward to receiving your revised manuscript.

Kind regards,

Arun Kumar Shanker

Academic Editor

PLOS ONE

Journal Requirements:

Additional Editor Comments:

The authors are requested to revise according to reviewer 2's comments

Reviewers' comments:

Reviewer's Responses to Questions

**Comments to the Author**

1. If the authors have adequately addressed your comments raised in a previous round of review and you feel that this manuscript is now acceptable for publication, you may indicate that here to bypass the “Comments to the Author” section, enter your conflict of interest statement in the “Confidential to Editor” section, and submit your "Accept" recommendation.

Reviewer #2: All comments have been addressed

2. Is the manuscript technically sound, and do the data support the conclusions?

Reviewer #2: Yes

3. Has the statistical analysis been performed appropriately and rigorously? 

Reviewer #2: Yes

4. Have the authors made all data underlying the findings in their manuscript fully available?

Reviewer #2: Yes

5. Is the manuscript presented in an intelligible fashion and written in standard English?

Reviewer #2: Yes

6. Review Comments to the Author

Reviewer #2: Using Paris polyphylla and arbuscular mycorrhiza fungi (AMF), the authors examined AMF colonization rate, relative water content, chlorophyll content and photosynthesis under drought stress. The experiments were designed well and executed efficiently. The methods and results are well justified towards the primary goal of this study. This study could be interesting to the Plant cell, Tissue and Organ Culture audience. A couple of minor things need to be solved before it is published.

1. In last sentence it is written that it is necessary to study the relationship between Dianthus, AMF and photosynthesis. The term Dianthus is not used anywhere. Is it Dianthus or P.Py.?

2. In materials and methods, how P.py seedlings were planted without 1 Kg of soil mixture. Sentence error?

7. PLOS authors have the option to publish the peer review history of their article (what does this mean?). If published, this will include your full peer review and any attached files.

Reviewer #2: No

---

## [Author Response · Author response to Decision Letter 1]

26 Sep 2023

The last references to Dianthus and without are errors of expression that have been corrected.

---

## [Decision Letter · Decision Letter 2]

2 Nov 2023

Mycorrhizal fungi reduce the photosystem damage caused by drought stress on Paris polyphylla var. yunnanensis

PONE-D-22-33646R2

Dear Dr. Huang

We’re pleased to inform you that your manuscript has been judged scientifically suitable for publication and will be formally accepted for publication once it meets all outstanding technical requirements.

Kind regards,

Arun Kumar Shanker

Academic Editor

PLOS ONE

Additional Editor Comments (optional):

The revision can be accepted as the authors have incorporated the suggestions of the reviewers.

Reviewers' comments:

Reviewer's Responses to Questions

**Comments to the Author**

1. If the authors have adequately addressed your comments raised in a previous round of review and you feel that this manuscript is now acceptable for publication, you may indicate that here to bypass the “Comments to the Author” section, enter your conflict of interest statement in the “Confidential to Editor” section, and submit your "Accept" recommendation.

Reviewer #2: All comments have been addressed

2. Is the manuscript technically sound, and do the data support the conclusions?

Reviewer #2: Yes

3. Has the statistical analysis been performed appropriately and rigorously? 

Reviewer #2: Yes

4. Have the authors made all data underlying the findings in their manuscript fully available?

Reviewer #2: Yes

5. Is the manuscript presented in an intelligible fashion and written in standard English?

Reviewer #2: Yes

6. Review Comments to the Author

Reviewer #2: Using Paris polyphylla and arbuscular mycorrhiza fungi (AMF), the authors examined AMF colonization rate, relative water content, chlorophyll content and photosynthesis under drought stress. The authors have underlined that the DS decreased the RWC and increased the OKJIP curve of the plants, and found that AMF inoculation could reverse it. This paper also reports the damage to the photosystem that the DS caused and demonstrates how the AMF assisted in mitigating the damage.The experiments were designed well and executed efficiently. The methods and results are well justified towards the primary goal of this study. This study could be interesting to the PLOS ONE.

7. PLOS authors have the option to publish the peer review history of their article (what does this mean?). If published, this will include your full peer review and any attached files.

Reviewer #2: No

---

## [Editor Report · Acceptance letter]

26 Mar 2024

PONE-D-22-33646R2 

PLOS ONE

Dear Dr. Huang, 

I'm pleased to inform you that your manuscript has been deemed suitable for publication in PLOS ONE. Congratulations! Your manuscript is now being handed over to our production team.

Kind regards, 

on behalf of

Dr. Arun Kumar Shanker 

Academic Editor

PLOS ONE